# Generative Inverse Design of Crystal Structures via Diffusion Models with Transformers

## Abstract

Recent advances in deep learning have enabled the generation of realistic data by training generative models on large datasets of text, images, and audio. While these models have demonstrated exceptional performance in generating novel and plausible data, it remains an open question whether they can effectively accelerate scientific discovery through the data generation and drive significant advancements across various scientific fields. In particular, the discovery of new inorganic materials poses a critical challenge, both scientifically and for industrial applications. However, unlike textual or image data, materials, or more specifically crystal structures, consist of multiple types of variables - including lattice vectors, atom positions, and atomic species. This complexity in data give rise to a variety of approaches for representing and generating such data. Consequently, the design choices of generative models for crystal structures remain an open question. In this study, we explore a new type of diffusion model for the generative inverse design of crystal structures, with a backbone based on a Transformer architecture. We demonstrate that our models are superior to previous methods in their versatility for generating crystal structures with desired properties. Furthermore, our empirical results suggest that the optimal conditioning methods vary depending on the dataset.

## 1 Introduction

The advancements in artificial intelligence, particularly in the domains of large language models and generative AI for image and audio synthesis, are having a significant impact on our social lives [OpenAI, 2023; Rombach *et al.*, 2022]. Such advancements in AI are also expected to accelerate research and development in materials science, which could potentially drive scientific discoveries and accelerate the development of materials. The discovery of novel materials, for example catalysis, battery materials, and superconducting materials, holds the potential to enable innovation in a wide range of industries [Toyao *et al.*, 2020; Chen *et al.*, 2020a].

Traditionally, the exploration of materials has required repeated try-and-error, consuming enormous amounts of time and effort. If novel and promising materials could be discovered in-silico, i.e., on computers, the exploration process could be further accelerated. Based on this concept, high-throughput virtual screenings using density functional theory (DFT) simulations or machine learning-based predictive models have been employed [Noh *et al.*, 2020]. However, such screening-based approaches have required enumerating and comprehensively simulating a vast number of candidate materials. If it were possible to selectively enumerate promising materials or directly generate materials with desired properties, the process of materials research and development would be significantly streamlined. To address these challenges, the inverse design of materials using deep generative models has emerged as a potential approach.

Materials, or more specifically crystal structures, consist of multiple types of variables including lattice vectors, atomic coordinates, and atomic species. There are several ways to represent crystal structures in a computational framework, and several approaches can be employed to generate them. Moreover, the inverse design requires generating crystal structures with desired properties rather than just generating them in a random manner, where several strategies can be employed to achieve this [Noh *et al.*, 2019; Xie *et al.*, 2021; Yang *et al.*, 2023; Zeni *et al.*, 2023]. In other words, there are various design choices for constructing the generative models for the inverse design of crystal structures.

Diffusion models are a type of generative model that have exhibited distinguished performance, particularly in the domain of image generation [Ho *et al.*, 2020; Song and Ermon, 2019]. They have also demonstrated its versatility and effectiveness in generating audio waveforms [Chen *et al.*, 2020b; Kong *et al.*, 2021], molecular and protein design [Hoogeboom *et al.*, 2022; Watson *et al.*, 2023], as well as crystal structure generation [Yang *et al.*, 2023; Jiao *et al.*, 2023; Zeni *et al.*, 2023]. In diffusion models, data is generated by progressively denoising an initial random input. For image generation tasks, variants of U-Net [Ronneberger *et al.*, 2015] based on convolutional neural networks (CNNs) has been used as the backbone for the denoising process. Recently, diffusion models replacing the U-Net with a Vision Transformers (ViTs) [Dosovitskiy *et al.*, 2021] have also been proposed, with the aim of improving scalability and enhanc-

ing the quality of generated data [Peebles and Xie, 2022; Hatamizadeh *et al.*, 2023]. This suggests that the examination of the backbone model in diffusion models can have a significant impact on the overall model performance.

In this study, we explore a new type of diffusion model for crystal structure generation, where the backbone is formulated based on a Transformer [Vaswani *et al.*, 2017] architecture. Furthermore, we explore suitable conditioning methods for the high-precision inverse design of crystal structures, and provide solid baselines for future research on generative crystal structure design techniques. Ultimately, we demonstrate that our proposed model is capable of performing inverse design with accuracy equal to or exceeding that of prior methods.

## 2 Preliminary

**Crystal structure** is characterized by the periodic arrangement of atoms in three-dimensional space. The crystal structure $M$ can be defined using a repeating unit called unit cell. When a unit cell contains $N$ atoms, the crystal structure $M$ can be represented as $M = (L, X, A)$, where $L = [l_1, l_2, l_3] \in \mathbb{R}^{3\times3}$ is the lattice matrix containing the lattice vectors of the unit cell, $X = [x_1, ..., x_N] \in \mathbb{R}^{3\times N}$ is the atomic coordinates in the Cartesian coordinate system, and $A = [a_1, ...a_N] \in \mathbb{Z}^N$ is the corresponding atomic species. $f_i = L^{-1}x_i \in [0, 1)^{3\times1}$ is called the fractional coordinate, which is advantageous for representing atomic positions in crystal structures considering their periodic nature. Therefore, the crystal structure can also be represented as $M = (L, F, A)$, where $F = [f_1, ..., f_N] \in [0, 1)^{3\times N}$ denotes the fractional coordinate matrix. A conceptual diagram of a crystal structure represented in two dimensions for simplicity is shown in Figure 1

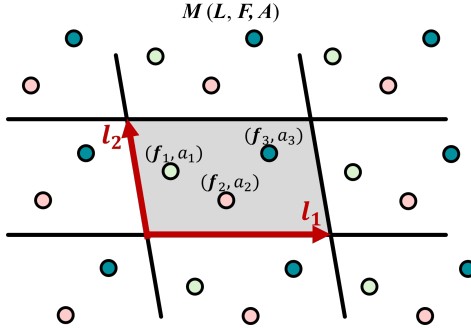

Figure 1: A conceptual diagram of a crystal structure represented in 2D for intuitive understanding. The red arrows represent the lattice vectors of the unit cell, while colored dots represent atoms, with the colors corresponding to different atomic species. The gray region highlights the area of the unit cell.

## 3 Related Work

### 3.1 Diffusion Models for Structured Data

Diffusion models are a class of generative models originally proposed for image generation, with two main formulation based on Denoising Diffusion Probabilistic Models (DDPMs) [Ho *et al.*, 2020] and Noise Conditional Score Networks (NCSCs) [Song and Ermon, 2019], where data is generated from the noise with the same dimension. Diffusion models have also been applied to other structured data, such as text [Hoogeboom *et al.*, 2021], point cloud [Luo and Hu, 2021] and graphs [Vignac *et al.*, 2023].

When applying to crystal structure generation, it is necessary to jointly model continuous variables, such as lattice vectors and atomic coordinates, alongside discrete variables, like atom types. Furthermore, to take into account the translational, rotational, and periodic invariance of crystal structures, it is necessary to consider the rotational equivariance of lattice vectors and the periodic invariance of fractional coordinates [Jiao *et al.*, 2023], which requires careful design of the denoising model architecture.

### 3.2 Transformers for Recognition and Generation

Transformer is a neural network architecture that was originally proposed for natural language processing (NLP) tasks [Vaswani *et al.*, 2017]. The self-attention mechanism in Transformers effectively captures long-range dependencies and efficiently encodes input sequences, leading to its success across various domains beyond NLP.

One prominent example of their application is in the field of computer vision. Vision Transformers (ViTs), which incorporate attention mechanisms to capture dependencies between image patches, have demonstrated superior performance compared to CNN-based models in image recognition tasks [Dosovitskiy *et al.*, 2021]. Due to this advantage, ViTs have recently been increasingly employed as backbone models for diffusion models in image generation tasks [Peebles and Xie, 2022; Gao *et al.*, 2023; Hatamizadeh *et al.*, 2023].

Furthermore, the application of Transformer-based models have advanced in domains related to materials science. For instance, self-attention mechanisms have been utilized to model the complex relationship between atoms within materials. This approach has enabled the effective representation learning of the 3D structures of molecules [Ying *et al.*, 2021] and materials [Yan *et al.*, 2022; Yan *et al.*, 2024; Taniai *et al.*, 2024], facilitating the accurate prediction of materials properties. Our study aims to leverage the power of self-attention mechanisms to model inter-atomic interactions, which is then used as the backbone of a diffusion model for crystal structure generation.

### 3.3 Generative Models for Crystal Generation

Generative models for crystal structures are essentially models designed to create representations of crystal structures $M = (L, F, A)$. In general, these models can be designed based on two aspects: how crystal structures are represented in a computational framework and how those representations are generated. Typical invertible crystal structure representations include 2D arrays containing crystallographic information on $M$ [Ren *et al.*, 2022], voxel images [Hoffmann *et al.*, 2019; Court *et al.*, 2020; Noh *et al.*, 2019], and graphs [Xie *et al.*, 2021; Luo *et al.*, 2023; Zeni *et al.*, 2023]. Regarding the generation methods, generative frameworks originally developed for image generation, such as Variational Autoencoders (VAEs) [Hoffmann *et al.*, 2019; Court *et al.*, 2020; Ren *et al.*, 2022; Xie *et al.*, 2021; Luo *et al.*, 2023; Noh *et al.*,

2019], Generative Adversarial Networks [Nouira *et al.*, 2018; Zhao *et al.*, 2021], and diffusion models [Xie *et al.*, 2021; Luo *et al.*, 2023; Zeni *et al.*, 2023; Yang *et al.*, 2023; Jiao *et al.*, 2023], have been widely adopted.

Many previous studies have focused on generating crystal structures for a limited range of crystal systems, such as cubic crystals, or for systems with a limited number of elements, such as binary or ternary compounds. However, recently, versatile generative models capable of generating diverse and plausible crystal structures across various crystal systems and elemental compositions have been developed. [Xie *et al.*, 2021; Luo *et al.*, 2023; AI4Science *et al.*, 2023; Yang *et al.*, 2023; Jiao *et al.*, 2023]. Xie *et al.* proposed CDVAE, a model that combines VAE and diffusion models, where crystal structures are represented as graphs. They demonstrated that the model is capable of generating diverse and reasonable crystal structures by learning the Cartesian coordinate scores using a graph neural network (GNN) as the backbone. Zeni *et al.* proposed a diffusion model that jointly generates $\boldsymbol{L}$, $\boldsymbol{X}$, and $\boldsymbol{A}$, which uses graphs as a representation of crystal structures, and the backbone model is based on a GNN [Zeni *et al.*, 2023]. A diffusion generative model by Yang *et al.* uses a well designed representation called Uni-Mat, and its backbone is based on a U-Net [Yang *et al.*, 2023].

In our diffusion model, atoms are treated as a point cloud in fractional space, and the backbone model is formulated based on a Transformer.

## 4 Methodology

In this section, we introduce our proposed model for the generative inverse design of crystal structures. We first introduce the joint diffusion framework in Sec. 4.1. In Sec. 4.2, an overview of the base architecture of the transformer model is provided, and in Sec. 4.3, two approaches of conditional generation for the generative inverse design of crystal structures are described.

### 4.1 Joint Diffusion Framework

As a general concept, a diffusion model defines two Markov processes: a fixed forward diffusion process that gradually adds noise to the original data $\boldsymbol{M}_0 = (\boldsymbol{L}_0, \boldsymbol{F}_0, \boldsymbol{A}_0)$ over $T$ steps, from $t = 1$ to $t = T$, and a learned generative process that removes the noise from the prior $\boldsymbol{M}_T = (\boldsymbol{L}_T, \boldsymbol{F}_T, \boldsymbol{A}_T)$.

**Diffusion on Lattices**

For lattice vectors with continuous variables, the forward diffusion process can be defined as follows [Jiao *et al.*, 2023], according to DDPM [Ho *et al.*, 2020]:

$$q(\boldsymbol{L}_t|\boldsymbol{L}_{t-1}) = \mathcal{N}(\boldsymbol{L}_t; \sqrt{1 - \beta_t}\boldsymbol{L}_{t-1}, \beta_t\mathbf{I}). \quad (1)$$

Here, $\beta_t \in [0, 1]$ is the predefined noise schedule, and $\mathbf{I}$ is the identity matrix. By applying the Markov property, $\boldsymbol{L}_t$ can be directly derived from $\boldsymbol{M}_0$ as:

$$q(\boldsymbol{L}_t|\boldsymbol{M}_0) = \mathcal{N}(\boldsymbol{L}_t; \sqrt{\overline{\alpha}_t}\boldsymbol{L}_0, (1 - \overline{\alpha}_t)\mathbf{I}), \quad (2)$$

where $\overline{\alpha}_t = \prod_{s=1}^{t} \alpha_s$ and $\alpha_t = 1 - \beta_t$. By the reparametrization trick, $\boldsymbol{L}_t$ can be written as $\boldsymbol{L}_t = \sqrt{\overline{\alpha}_t}\boldsymbol{L}_0 + \sqrt{1 - \overline{\alpha}_t}\boldsymbol{\epsilon}_L$, where $\boldsymbol{\epsilon}_L \sim \mathcal{N}(\mathbf{0}, \mathbf{I})$.

In the backward process, the lattice vectors are represented using a Gaussian distribution $\mathcal{N}(\boldsymbol{L}_{t-1}|\boldsymbol{\mu}_\theta(\boldsymbol{M}_t), \Sigma_t\mathbf{I})$, where $\boldsymbol{\mu}_\theta(\boldsymbol{M}_t) = \frac{1}{\sqrt{\alpha_t}}(\boldsymbol{L}_t - \frac{\beta_t}{\sqrt{1 - \overline{\alpha}_t}}\hat{\boldsymbol{\epsilon}}_L(\boldsymbol{M}_t, t))$ and $\Sigma_t = \beta_t\frac{1 - \overline{\alpha}_{t-1}}{1 - \overline{\alpha}_t}$. The neural network is trained to predict $\hat{\boldsymbol{\epsilon}}_L$, given $\boldsymbol{M}_t$ and $t$, with the loss function

$$\mathcal{L}_L = \mathbb{E}_{\boldsymbol{\epsilon}_L \sim \mathcal{N}(\mathbf{0}, \mathbf{I}), t \sim \mathcal{U}(1, T)}[||\boldsymbol{\epsilon}_L - \hat{\boldsymbol{\epsilon}}_L(\boldsymbol{M}_t, t)||_2^2]. \quad (3)$$

**Diffusion on Coordinates**

While fractional coordinates $\boldsymbol{F} \in [0, 1)^{3 \times N}$ are convenient for handling periodicity, applying DDPM with Gaussian functions is not appropriate. Therefore, score-matching based models [Song and Ermon, 2019; Song *et al.*, 2019; Song and Ermon, 2020] have been adopted together with wrapped normal (WN) distribution [Jiao *et al.*, 2023; Zeni *et al.*, 2023]. In this case, we assume that $\boldsymbol{F}_t$ follows the perturbed distribution with a predefined noise schedule $\sigma_t$ as follows:

$$q(\boldsymbol{F}_t|\boldsymbol{F}_0) = \mathcal{N}_w(\boldsymbol{F}_t; \boldsymbol{F}_0, \sigma_t^2\mathbf{I}), \quad (4)$$

where $\mathcal{N}_w$ denotes a WN distribution, and $\sigma_t$ is defined using a hyper parameter $\sigma_T$ as $\sigma_0 = 0$ and $\sigma_t = \sigma_1(\frac{\sigma_T}{\sigma_1})^{\frac{t-1}{T-1}}$ for $t > 0$.

In the backward process, the output of neural network $\hat{\boldsymbol{\epsilon}}_F$ estimates the score of perturbed data distribution, where the neural network is trained with the objective:

$$\mathcal{L}_F = \mathbb{E}_{\boldsymbol{F}_t \sim q(\boldsymbol{F}_t|\boldsymbol{F}_0), t \sim \mathcal{U}(1, T)}[\lambda_t||\nabla_{\boldsymbol{F}_t} \log q(\boldsymbol{F}_t|\boldsymbol{F}_0) - \hat{\boldsymbol{\epsilon}}_F(\boldsymbol{M}_t, t)||_2^2]. \quad (5)$$

Here, $\lambda_t = \mathbb{E}_{\boldsymbol{F}_t}^{-1}[||\nabla_{\boldsymbol{F}_t} \log q(\boldsymbol{F}_t|\boldsymbol{F}_0)||_2^2]$ is approximated via Monte Carlo Sampling, and the data is generated by the ancestral predictor with the Langevin corrector, as detailed in [Jiao *et al.*, 2023].

**Diffusion on Species**

There are several possible approaches to modelling the diffusion of atomic species; however, in this study, we consider them as categorical data and apply discrete denoising diffusion probabilistic models (D3PMs) [Austin *et al.*, 2023; Hoogeboom *et al.*, 2021], as employed in the previous works [Guan *et al.*, 2023; Peng *et al.*, 2023; Zeni *et al.*, 2023].

In D3PM, the forward diffusion process is formulated using categorical distribution with probability vector $\boldsymbol{p}$ as:

$$q(\boldsymbol{a}_{i,t}|\boldsymbol{a}_{i,t-1}) = \mathrm{Cat}(\boldsymbol{a}_{i,t}; \boldsymbol{p} = \boldsymbol{a}_{i,t-1}\boldsymbol{Q}_t), \quad (6)$$

where $\boldsymbol{a}_{i,t} \in \{0, 1\}^K$ is the one-hot row-vector representation of atomic species $a_i$ at timestep $t$, $\boldsymbol{Q}_t$ is the transition matrix, and $K$ is the number of classes. The Markov property allows for directly computing $\boldsymbol{a}_{i,t}$ from $\boldsymbol{a}_{i,0}$ as follows:

$$q(\boldsymbol{a}_{i,t}|\boldsymbol{a}_{i,0}) = \mathrm{Cat}(\boldsymbol{a}_{i,t}; \boldsymbol{p} = \boldsymbol{a}_{i,0}\overline{\boldsymbol{Q}}_{i,t}), \quad (7)$$

with $\overline{\boldsymbol{Q}}_t = \boldsymbol{Q}_1\boldsymbol{Q}_2...\boldsymbol{Q}_t$. In this study, we define transition matrix as $\boldsymbol{Q}_t = (1 - \beta_t)\mathbf{I} + \beta_t/K\mathbb{1}\mathbb{1}^\top$ so that atomic species follow a uniform distribution at $t = T$, which was selected based on experiments.

The backward process is modeled using categorical distribution $\mathrm{Cat}(\boldsymbol{a}_{i,t-1}; \boldsymbol{p} = \frac{\boldsymbol{a}_{i,t}\boldsymbol{Q}_t^\top \odot \hat{\boldsymbol{a}}_{i,0}\overline{\boldsymbol{Q}}_{t-1}}{\hat{\boldsymbol{a}}_{i,0}\overline{\boldsymbol{Q}}_t\boldsymbol{a}_{i,t}^\top})$, wherein $\hat{\boldsymbol{a}}_{i,0} = \hat{\boldsymbol{a}}_{i,0}(\boldsymbol{M}_t, t)$ is predicted by the neural network, which is trained with the loss function:

$$\mathcal{L}_a = \mathbb{E}_{\boldsymbol{a}_{i,t} \sim q(\boldsymbol{a}_{i,t}|\boldsymbol{a}_{i,0}), t \sim \mathcal{U}(1, T)}[q(\boldsymbol{a}_{i,t-1}|\boldsymbol{M}_t, \boldsymbol{M}_0)||p_\theta(\boldsymbol{a}_{i,t-1}|\boldsymbol{M}_t)]. \quad (8)$$

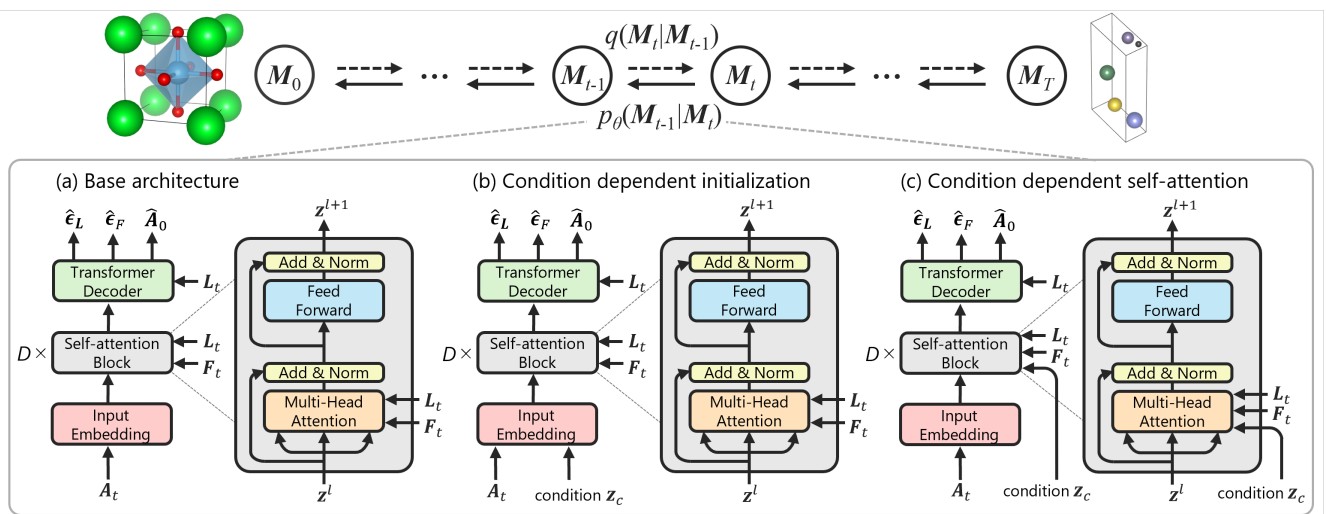

Figure 2: (a) Overview of the base model architecture. (b) Overview of the conditional model with condition-dependent initialization method. (c) Overview of the conditional model with condition-dependent self-attention method. The models of crystal structures in this figure were visualized using VESTA [Momma and Izumi, 2008].

## 4.2 Backbone Denoising Model

The overview of the base model architecture is shown in Figure 2 (a). Under the joint diffusion processes defined in Sec. 4.1, we construct a model that receives $M_t = (L_t, F_t, A_t)$ as input and outputs $\hat{\epsilon}_L$, $\hat{\epsilon}_F$, and $\hat{A}_0$. In the following, we describe several key components of the model.

**Input Embedding**

As shown in Figure 2, initial input embedding in a unit cell, $z^0 = (z_1^0, ..., z_N^0)$, which is fed into the self-attention block, is generated based on time-dependent atomic species $A_t = (a_{1,t}, ..., a_{N,t})$. We generate $d$-dimensional species embedding $z_{a_i} = \text{MLP}(a_{i,t}) \in \mathbb{R}^d$ and set $z_i^0 = z_{a_i}$, where MLP denotes a simple multi-layer perceptron.

**Self-Attention Block**

The Self-Attention block, which serves as a core component of the proposed model, receives $z^l = (z_0^l, ..., z_N^l)$, $L_t$, and $F_t$ as inputs, and outputs an updated $z^{l+1} = (z_0^{l+1}, ..., z_N^{l+1})$, where $d$ denotes block index $l = 0, ..., D-1$. Following the original Transformer [Vaswani et al., 2017], $z^l$ is updated as follows:

$$\hat{z} = \text{LN}(\text{MHA}(z^l, L_t, F_t) + z^l), \tag{9}$$

$$z^{l+1} = \text{LN}(\text{MLP}(\hat{z}) + \hat{z}), \tag{10}$$

where MHA and LN denote multi-head attention layer and layer normalization [Ba et al., 2016], respectively.

In this work, the attention mechanism in the MHA layer relies on self-attention with relative position representations [Shaw et al., 2018] to better incorporate relative positional relationships between atoms. Specifically, input sequences $z = (z_0, ..., z_N)$ are transformed to $z' = (z_0', ..., z_N')$ according to the following equation:

$$z_i' = \frac{1}{Z_i} \sum_{j=1}^{N} \exp(q_i^\top k_j / \sqrt{d} + \alpha_{ij})(v_j + \beta_{ij}). \tag{11}$$

Here, $Z_i = \sum_{j=1}^{N} \exp(q_i^\top k_j / \sqrt{d} + \alpha_{ij})$, and $k_i = z_i W^K$, $q_i = z_i W^Q$, $v_i = z_i W^V$, where $W^Q$, $W^K$, $W^V$ are parameter matrices for query, key, and value, respectively.

The terms $\alpha_{ij} \in \mathbb{R}$ and $\beta_{ij} \in \mathbb{R}^d$ serve as biases to incorporate the relative positional relationship between atoms $i$ and $j$. In this study, we consider generating these terms from $F_t$ and $L_t$. Specifically, to incorporate information about the relative positions of atom $i$ and $j$ in fractional space, a Fourier transformation $\psi_{\text{FT}}(f_j - f_i)$, which is first proposed in [Jiao et al., 2023] for creating periodic-invariant message in GNN, is utilized alongside $L_t$ to get $\alpha_{ij}$ and $\beta_{ij}$ as follows:

$$\alpha_{ij} = \text{MLP}(\psi_{\text{FT}}(f_j - f_i), L_t^\top L_t), \tag{12}$$

$$\beta_{ij} = \text{MLP}(\psi_{\text{FT}}(f_j - f_i), L_t^\top L_t). \tag{13}$$

Here, the $\psi_{\text{FT}}$ is defined as:

$$\psi_{\text{FT}}(f)[n, k] = \begin{cases} \sin(2\pi m f_n) & \text{if } k = 2m \\ \cos(2\pi m f_n) & \text{if } k = 2m+1, \end{cases} \tag{14}$$

where $n$ is an index that runs over the dimension of 3D coordinates, and $k$ is an index that runs over the dimensions of the embedding vector. In plactice, $\psi_{\text{FT}}(f_j - f_i)$ and $L_t^\top L_t$ are flattened and concatenated before being fed into MLP.

**Transformer Decoder**

After updating through the $D$-layer self-attention block, the transformer decoder receives $z^D = (z_0^D, ..., z_N^D)$ and $L_t$, and then predicts $\hat{\epsilon}_L$, $\hat{\epsilon}_F$, and $\hat{A}_0$. $\hat{\epsilon}_F$ and $\hat{A}_0$ are obtained straightforwardly from $z^D$ through the equations $\hat{\epsilon}_{f_i} = \text{MLP}(z_i^D)$ and $\hat{a}_{i,0} = \text{MLP}(z_i^D)$. $\hat{\epsilon}_L$ is obtained by the linear transformation of $L_t$ as $\hat{\epsilon}_L = L_t \varphi(\frac{1}{N} \sum_{i=1}^{N} z_i^D)$, where $\varphi$ denote an MLP that outputs a $3 \times 3$ matrix. We note that this procedure follows the steps performed in the read-out of GNN [Jiao et al., 2023], and enables prediction of $\hat{\epsilon}_L$ that is equivariant to the rotation of $L_t$.

| Method | Perov-5 | | | Carbon-24 | | | MP-20 | | |
|---|---|---|---|---|---|---|---|---|---|
| | SR5 | SR10 | SR15 | SR5 | SR10 | SR15 | SR5 | SR10 | SR15 |
| CDVAE [Xie *et al.*, 2021] | 0.52 | 0.65 | 0.79 | 0.00 | 0.06 | 0.06 | 0.78 | 0.86 | 0.90 |
| SyMat [Luo *et al.*, 2023] | 0.73 | 0.80 | 0.87 | 0.06 | 0.13 | 0.13 | **0.92** | **0.97** | **0.97** |
| MODEL-CDI | 0.93 | 0.97 | 0.98 | 0.43 | 0.56 | 0.57 | 0.91 | 0.93 | 0.95 |
| MODEL-CDS | **0.97** | **0.99** | **1.00** | **0.56** | **0.64** | **0.64** | 0.63 | 0.75 | 0.84 |

Table 1: Property optimization performance, where SR stands for success rate. The highest SR and the second highest SR achieved among the four models are emphasized with bold and underline, respectively. Performance of CDVAE and SyMat were obtained from the original literatures.

## 4.3 Conditioning Methods

In this study, we perform conditional generation, using physical property values as condition, for the generative inverse design of crystal structures. We provide the model with the time step $t$ and property values as conditions, where time step $t$ is transformed into a feature vector $z_t$ using sinusoidal positional encoding [Vaswani *et al.*, 2017; Ho *et al.*, 2020], while property values are mapped to a vector $z_{\text{prop}}$ via linear projection. The condition embedding is constructed from $z_t$ and $z_{\text{prop}}$ as follows:

$$z_c = \text{MLP}(z_t \oplus z_{\text{prop}}), \tag{15}$$

where $\oplus$ denotes the concatenation of two vectors. As described below, we investigate two conditional generation approaches based on how $z_c$ is fed into the model.

### Condition Dependent Initialization (CDI)

The first approach is to make the input embedding condition-dependent, as shown in Figure 2 (b). In this case, input embedding is set using not only the species embedding $z_{a_i}$ but also the condition embedding $z_c$ as follows:

$$z_i^0 = \text{MLP}(z_{a_i} \oplus z_c). \tag{16}$$

The model conditioned with condition-dependent initialization is referred to as MODEL-CDI.

### Condition Dependent Self-attention (CDS)

Figure 2 (c) shows the second approach, where $z_c$ is fed into every self-attention block. Similar to the Time-dependent Self-attention mechanism proposed in DiffiT [Hatamizadeh *et al.*, 2023], the attention block takes $z_c$ as an additional input, and the query, key, value are made condition-dependent as follows:

$$q_i = z_i W^Q + z_c W^{Qc}, \tag{17}$$
$$k_i = z_i W^K + z_c W^{Kc}, \tag{18}$$
$$v_i = z_i W^V + z_c W^{Vc}, \tag{19}$$

where $W^{Qc}$, $W^{Kc}$, $W^{Vc}$ are the conditional projection matrices for query, key, and value, respectively. The model conditioned with this condition-dependent self-attention is referred to as MODEL-CDS.

## 5 Experiments

In this section, we evaluate the performance of our proposed models on the task of property optimization. Through the comparison with existing methods, we show that our proposed models are effective for generative inverse design of crystal structures.

### 5.1 Experimental Setup

**Task definition**

Property optimization is a task that aims to generate crystal structures that possess desired physical properties when target property values are provided.

**Datasets**

To evaluate the performance of proposed models across diverse materials systems, we conduct the assessments on three datasets with different compositions and crystal systems, following Xie *et al.* [Xie *et al.*, 2021]. **Perov-5** [Castelli *et al.*, 2012a; Castelli *et al.*, 2012b] is a dataset of crystal structures derived from cubic perovskite, containing 18,928 crystal structures and their corresponding property values. **Carbon-24** [Pickard, 2020] contains 10,153 structures consisting solely of carbon atoms, with each structure including between 6 and 24 atoms. **MP-20** [Jain *et al.*, 2013] is a collection of stable crystal structures from the Materials Project database, each containing 20 atoms or fewer. MP-20 includes a total of 45,231 structures and and is the most diverse dataset in terms of both composition and crystal systems. Following [Xie *et al.*, 2021], each dataset was split into a ratio of 6:2:2 for training, validation, and testing, respectively.

**Metrics**

In the task of property optimization, we measure the ability of generative models to generate crystal structures with low formation energies, in other words, to generate stable crystal structures. In this research, we perform conditional generation of crystal structures by providing the minimum values of the physical properties in the training dataset as conditional values, and calculate the success rate (SR). The success rate is calculated as the proportion of generated crystal structures whose physical property values fall within the top 5% (SR5), 10% (SR10), and 15% (SR15) of the target values. The property values were calculated using pre-trained GNN model by Xie *et al.* [Xie *et al.*, 2021].

**Baselines**

We compare our models with two existing methods: CD-VAE [Xie *et al.*, 2021] and SyMat [Luo *et al.*, 2023], both of which are crystal structure generation models that combine VAE and score-based diffusion models, and they can be applied to the property optimization. Other diffusion models, such as MatterGen [Zeni *et al.*, 2023] and UniMat [Yang *et al.*, 2023], were not compared because source codes are not open at present and were not assessed on the same metrics.

## 5.2 Results

The performance of property optimization is reported in Table 1. For both Perov-5 and Carbon-24, our proposed models (MODEL-CDI and MODEL-CDS) demonstrated superior performance compared to the previous methods (CDVAE and SyMat). Notably, for Perov-5, MODEL-CDS yielded SRs of nearly 1.0, while for Carbon-24, the SR5, which was close to 0.0 with previous methods, increased to above 0.5. On the other hand, for MP-20, using MODEL-CDI achieved SRs as high as that of SyMat, while MODEL-CDS resulted in a slightly lower SRs. The fact that the proposed models demonstrated overall good performance suggests that Transformers can also be applied as the backbone in diffusion models for crystal structure generation. In addition, as shown in Table 1, MODEL-CDI demonstrated SRs close to the best across all datasets. This indicates that MODEL-CDI is a versatile model capable of being applied to the task of generating crystal structures with desired properties across a wide variety of datasets.

It is interesting to note that when comparing the performance of MODEL-CDI and MODEL-CDS, MODEL-CDS demonstrated higher performance on Perov-5 and Carbon-24, while MODEL-CDI exhibited better performance on MP-20. From the perspective of the crystal structure distribution, crystal structures in Perov-5 has a high degree of freedom in composition, while that in Carbon-24 has a high degree of freedom in crystal systems. MP-20, on the other hand, has high degrees of freedom in both composition and crystal systems. Therefore, it is conceivable that the factors determining the property values, particularly the formation energy in this case, vary depending on the dataset. The difference in performance between MODEL-CDI and MODEL-CDS in this study is considered to reflect this differences of characteristics of datasets, indicating that the optimal conditioning method varies depending on the dataset. The optimal method for conditional generation may also vary depending on the model architecture and the target properties. Therefore, exploring suitable conditional generation techniques is expected to be a valuable direction for future research.

## 6 Conclusion and Future Work

In this work, we explored a new diffusion model for generative inverse design of crystal structures, where the backbone is formulated based on a Transformer architecture. We explored two conditioning methods for generating crystal structures with target physical properties. Our models generally demonstrated comparable or superior performance compared to previous methods. Furthermore, it was found that the optimal conditioning method varies depending on the dataset, suggesting that the exploration of conditioning techniques depending on the dataset and property would be important for high-precision inverse design.

As future work, evaluation using DFT calculations will be necessary for more rigorous assessments. Additionally, Additionally, the novelty and diversity of the inversely designed crystal structures will also become important metrics in the discovery of new materials.

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
