# OpenReview forum: "Generative Inverse Design of Crystal Structures via Diffusion Models with Transformers"
_ijcai.org/IJCAI/2024/Workshop/AI4Research — AI4Research 2024_

### Official Review · Reviewer_bLqh · 2024-05-31
**Marginally below acceptance threshold (weak reject)**

**Rating:** 5
**Confidence:** 4

**Review:**

The paper proposed a transformer-based diffusion method for crystal structure design. I think the advantages and disadvantages of the paper are obvious. The main problem is that the experiments are not enough to fully demonstrate the performance of the method. The advantages of the paper are:
1. The methods generate the crystal structures with a diffusion model from 3 different perspectives with a transformer backbone model.
2. investigating the physical properties as conditionals to enhance the method

However, on the other hand, the disadvantages are more obvious. For this paper, I think the disadvantages are:
1. The most significant disadvantage is the experiments and this is the main reason I say the paper blows the acceptance threshold. The paper only performs one experiment focusing on property optimization and compares with only 2 baselines. From other related papers, I found that they also do experiments to evaluate material generation. Based on the introduction and other sections, I think the paper should also consider the material generation evaluation. If the scope of this paper is just focusing on property optimization, it should be mentioned in the sections such as preliminary or problem statement. Another thing is that the experiments only compare with two diffusion baselines, which is too limited. In the introduction and related works, the authors mentioned many other works that use models such as VAE and GAN. I think these baselines should also be considered to be compared. At least using 4 baselines. Moreover, in this work, there is a base model without any condition cases. Why this base model is not evaluated?
2. The motivation is not clear. Why do you want to use a transformer as a backbone model? What challenges drive you to use a transformer-based diffusion model? What are the differences between the transformer model with others? Why transformer better than other backbone models? Please briefly clarify these motivations in the introduction section.

---

### Official Review · Reviewer_oLKv · 2024-06-01
**Generative Inverse Design of Crystal Structures via Diffusion Models with Transformers**

**Rating:** 6
**Confidence:** 3

**Review:**

This paper introduces a joint diffusion framework that encompasses both continuous and discrete variables, such as lattice vectors, atomic coordinates, and types of atoms. This framework is applied in the field of materials science, particularly for the inverse design of crystal structures, enabling more effective generation of crystal structures with desired properties.

Advantages:

1. The method uses a diffusion model and transformer architecture for the inverse design of crystal structures, representing an innovation in the field of materials science.
2. Compared to existing methods, this study achieves a higher success rate in property optimization tasks across various datasets.

Questions:

1. Could you provide more details about the transformer architecture used in the diffusion model, such as the dimensions of the layers and the training configurations?
2. Could you provide additional ablation study results or analyses to demonstrate the impact of each component in the model, particularly the newly introduced conditioning methods (CDI and CDS)?
3. How does the model's performance vary with different types of crystal structures and compositions? Are there specific features of the datasets that significantly affect the effectiveness of the proposed model?
4.How did you choose the benchmarks for comparison? Are there more extensive and comprehensive benchmarks available?

---

### Decision · Program_Chairs · 2024-06-03

Accept